# Temperature-Constrained LiDAR Retrieval of Planetary Boundary Layer Height over Chiang Mai, Thailand

Ronald Macatangay[1], Thiranan Sonkaew[2], Sherin Hassan Bran[1], Worapop Thongsame[3], Titaporn Supasri[1], Mana Panya[1*], Jeerasak Longmali[1], Raman Solanki[4**], Ben Svasti Thomson[5], Achim Haug[6]

[1]National Astronomical Research Institute of Thailand, Don Kaeo, Mae Rim District, Chiang Mai 50180, Thailand
[2]Science Faculty, Lampang Rajabhat University, Chomphu, Mueang, Lampang 52100, Thailand
[3]University of Colorado, Boulder, CO 80309, USA
[4]Indian Institute of Tropical Meteorology (IITM), Pune, India
[5]Blue Sky Chiang Mai, Chiang Mai, Thailand
[6]Air Gradient, Chiang Mai, Thailand

*Correspondence to*: Ronald Macatangay (ronmcdo@gmail.com / ronald@narit.or.th) / Thiranan Sonkaew (thiranan.sonkaew@gmail.com / thiranan@g.lpru.ac.th)

*now at the Department of Highland Agriculture and Natural Resources / Agriculture and Forestry Climate Change Research Center (AFCC), Faculty of Agriculture, Chiang Mai University 50200, Thailand

**currently an independent researcher

**Abstract.** Accurate determination of the planetary boundary layer (PBL) height, mixing layer height (MLH), and aerosol layer top (ALT) is essential for air quality and climate studies, particularly in regions with complex aerosol dynamics such as Chiang Mai, northern Thailand. This study introduces a novel LiDAR-based retrieval method that integrates a temperature-dependent, dynamic maximum analysis altitude (MAA) into the traditional Haar Wavelet Covariance Transform (WCT) framework. Unlike conventional fixed-altitude WCT approaches, which often misclassify the ALT as the PBL—especially under stable nighttime or transitional conditions—this dynamic approach adapts the vertical search range for PBL detection in real time using observed surface temperature variations. The method is physically grounded in boundary layer thermodynamics, allowing for more accurate identification of the true PBL top while reducing contamination from residual aerosol layers and low clouds. Validation against radiosonde observations and comparison with previously validated WRF-Chem simulations demonstrate strong agreement, with the LiDAR-derived PBL heights capturing diurnal variations more reliably than traditional methods. The findings also reveal model biases during high aerosol events, highlighting the need for improved aerosol–meteorology coupling in mesoscale models. This integrated retrieval framework represents a significant advancement in LiDAR-based boundary layer detection and offers a robust tool for enhancing pollutant dispersion analysis, air quality forecasting, and climate modeling across aerosol-rich regions in Southeast Asia.

## 1 Introduction

The planetary boundary layer (PBL) height, mixing layer height (MLH), and aerosol layer top (ALT) are distinct atmospheric parameters essential for understanding air quality and weather dynamics. The PBL height represents the top of the lowest atmospheric layer influenced by surface interactions, encompassing layers such as the daytime mixing layer and the stable boundary layer at night (Stull, 1988). The MLH, a turbulent and well-mixed subset of the PBL, typically aligns with the PBL height under convective conditions but can diverge in stratified or stable atmospheres (Seibert et al., 2000). The ALT, marking the upper boundary of significant aerosol concentrations, often decouples from the PBL due to stratification, entrainment, or advection processes. Moreover, due to stronger vertical mixing over mountainous topography the aerosol can reach higher altitudes often resulting in significant aerosol loading above the PBL (De Wekker and Kossmann, 2015). Light Detection and Ranging (LiDAR) based techniques, despite offering high-resolution aerosol profiles, frequently face challenges in distinguishing these layers, particularly during nighttime or transitions (Ferrare et al., 2012). Emerging approaches that combine normalized relative backscatter (NRB) data with thermodynamic adjustments address these challenges, improving the accuracy of boundary layer height and ALT identification (Su et al., 2018; NASA DISCOVER-AQ Workshop, 2012). These developments are particularly relevant for regions like Southeast Asia, where high aerosol loads from industrial and biomass burning activities create complex vertical profiles.

This study focuses on enhancing LiDAR-based boundary layer characterization by refining the detection of PBL height, MLH, and ALT. Traditional algorithms, such as the Haar Wavelet Covariance Transform (WCT), often misclassify the ALT as the PBL height, especially at night or during transitional periods when aerosol gradients are less distinct. Clouds and other atmospheric complexities make these measurements more challenging. By integrating normalized relative backscatter (NRB) profiles with dynamic thermodynamic adjustments, this approach addresses ambiguities in traditional methods and improves the reliability of boundary layer determinations. The novel method developed in this study was validated using radiosonde measurements and compared against WRF-Chem simulations. To support model–observation comparisons, we use a WRF-Chem configuration that has been previously validated under similar regional conditions in northern Thailand for surface pollutant distributions, boundary layer dynamics, and optical turbulence (Bran et al., 2022; Macatangay et al., 2024; Bran et al., 2024), confirming its suitability as a benchmark.

Numerous studies have significantly advanced the methods for estimating planetary boundary layer (PBL) height using LiDAR, addressing challenges posed by complex meteorological conditions, aerosol stratification, and limitations in traditional methodologies. For example, Toledo et al. (2017) explored numerical methods under sea–land breeze regimes, revealing discrepancies in residual layers that affect accurate PBL height detection, while Vishnu et al. (2017) highlighted the difficulty of applying a universal method for estimating the mixing layer height (MLH), which varies under different atmospheric conditions. Li et al. (2017) contributed to improving convective boundary layer height (CBLH) retrievals by developing a convective condensation level algorithm, and Dang et al. (2019) reviewed various aerosol LiDAR techniques and

developed a robust method (Dang et al., 2019), achieving strong correlations with radiosonde data to enhance accuracy. In addition, Zhong et al.'s (2020) MLHI-RR technique, Zhang et al.'s (2020) Cluster Analysis of the Gradient Method, and Macatangay et al.'s (2021) TDMMAA method for Haar wavelet techniques improved the precision of PBL measurements, while machine learning approaches, such as Liu et al.'s (2022) MKnm algorithm, and other methods like Pan et al.'s (2022) MR-IP and Han et al.'s (2022) ADEILP, have addressed multilayer conditions and diurnal variations in aerosol profiles.

A critical challenge remains in fully distinguishing the PBL height, MLH, and ALT, particularly in regions with high aerosol variability, such as Northern Thailand. Diverse pollution sources, including biomass burning, anthropogenic, and biogenic emissions, contribute to intricate vertical aerosol distributions. This study attempts to overcome the limitations of WCT, improving the separation of PBL, MLH, and ALT through a novel integrated NRB and surface temperature approach. The

enhanced accuracy of boundary layer detection is pivotal for improving air quality and mesoscale meteorological models, particularly in Southeast Asia, where aerosol concentrations are highly variable and challenging to characterize. This work provides new insights into the vertical aerosol distribution and its implications for air quality, aerosol-meteorology interactions, and public health in regions heavily impacted by agricultural and industrial emissions.

## 2 Methodology

To estimate the planetary boundary layer (PBL) height using micropulse LiDAR, normalized relative backscatter (NRB) profiles were analyzed, as NRB is proportional to aerosol concentration (Campbell et al., 2002). Conventional methods, such as the Haar WCT with a fixed maximum analysis altitude (e.g., 4 km), often yield the top of the aerosol layer rather than the actual PBL height. These methods can overestimate the PBL height during nighttime by identifying the residual layer top, and during transitional periods (morning growth or evening decay), they fail to capture diurnal PBL variations accurately (Brooks

et al., 2003). Moreover, low clouds can lead to incorrect identification of either their base or top as the PBL height. To address these limitations, a robust algorithm that adapts dynamically to atmospheric conditions is necessary.

The study was conducted at the headquarters of the National Astronomical Research Institute of Thailand (NARIT), situated at the Princess Sirindhorn AstroPark in Chiang Mai, northern Thailand (18.85° N, 98.96° E, 332 mASL), a region known for its complex aerosol dynamics driven by mountainous topography, biomass burning (forest and agricultural fires),

anthropogenic pollution, and biogenic emissions from forested areas. The site experiences significant seasonal variations in aerosol concentrations, influencing the vertical distribution of particulate matter. To monitor these dynamics, a Mini Micro Pulse LiDAR (MiniMPL) system was deployed at the site and remained fixed throughout the measurement campaign to ensure spatial consistency. The instrument operated at a pulse repetition frequency of 2500 Hz, with each vertical profile averaged over 30 seconds. For analysis, the data were grouped into 5-minute intervals to balance temporal resolution with noise

reduction, in line with standard practices in boundary layer studies, including those at this site (Solanki et al., 2019). The LiDAR was continuously pointed at the zenith, ensuring consistent spatiotemporal overlap for robust monitoring of the vertical

aerosol structure and planetary boundary layer (PBL) evolution. The study period, from December 2023 to February 2024, coincides with the beginning of the dry season in northern Thailand, when agricultural and forest fires begin to elevate aerosol concentrations (Bran et al., 2024), complicating the identification of the PBL and aerosol layers. This period provides a unique opportunity to evaluate the performance of the proposed LiDAR-based approach under high aerosol loading and variable meteorological conditions. A more comprehensive description of the study site is given in Solanki et al., 2019.

The dynamic maximum analysis altitude (MAA) method introduced in this study is grounded in boundary layer thermodynamics and turbulence theory. It addresses a key limitation in conventional planetary boundary layer (PBL) detection approaches — namely, the use of a fixed maximum altitude for analysis regardless of prevailing atmospheric conditions. By leveraging real-time surface temperature variations, this method introduces a thermodynamically responsive upper boundary for LiDAR-based boundary layer retrievals. The physical rationale stems from the well-established relationship between surface heating, buoyant turbulence generation, and boundary layer growth. Under convective conditions, surface warming leads to rising thermals that entrain air and deepen the boundary layer (Stull, 1988). Conversely, cooler surface temperatures typically indicate stable stratification or residual layer conditions in the early morning or late evening (Seibert et al., 2000). These thermal variations strongly influence the height and structure of the PBL, as described in classic boundary layer turbulence models such as the Mixed Layer Model (Tennekes, 1973; Garratt, 1994) and first-order closure turbulence schemes implemented in models like WRF-Chem (Skamarock et al., 2008). This dynamic parameter, unlike the fixed altitudes used in conventional methods, is calculated using Equation (1):

$$MAA(t) = LAA + (HAA - LAA)\left(\frac{T(t) - min(T)}{max[T - min(T)]}\right) \qquad (1)$$

where MAA(t) is the time-varying surface temperature-based maximum analysis altitude

LAA and HAA, represent the lowest and highest allowable maximum analysis altitudes. These are set to 0.5 and 2.5 km, respectively, based on Solanki et al., 2019

T(t) is the observed surface temperature (in °C)

min(T) is the minimum temperature of the day (or the previous day for operational use)

max[ ] is the maximum of the expression inside the brackets

t is time, representing temporal variation for all T, with data recorded every 5 minutes. The normalization ensures that MAA is low under cooler conditions (e.g., early morning residual layer regimes) and higher under warmer, convectively unstable conditions typical of late morning and afternoon boundary layer growth. This dynamic framework enhances robustness when detecting the PBL top using the Haar wavelet covariance transform (WCT) method (Brooks, 2003), as implemented in the

Ceilometer Layer Identification and Optimization (Ceilo) code. Following the WCT detection, a 6-hour moving average is applied to the raw PBL height time series to suppress high-frequency variability associated with short-lived turbulence bursts or instrumental noise. The resulting PBL height estimates are subsequently validated against radiosonde measurements and WRF-Chem model outputs to assess performance and reproducibility. This methodology contributes to a growing body of literature advocating adaptive and physically-informed PBL detection methods (Hennemuth & Lammert, 2006), particularly under complex aerosol and meteorological regimes like those encountered in Southeast Asia.

Data from radiosondes launched by the Thai Meteorological Department (TMD) at the Chiang Mai International Airport (18.77° N, 98.96° E, 311 mASL; approximately 9 km in distance from the study site) were retrieved from the University of Wyoming's atmospheric sounding archive (https://weather.uwyo.edu/upperair/sounding.html). These data were interpolated to a vertical grid with 30-meter spacing from 100 m to 2 km, corresponding to the LiDAR minimum detection height or overlap region and the typical aerosol layer top (Solanki et al., 2019), respectively. Although the radiosonde data were interpolated only up to 2 km for consistency with the LiDAR's effective detection range and overlap region, the dynamic maximum analysis altitude (MAA) used in our retrieval method was permitted to extend up to 2.5 km. This was done to allow for the detection of elevated convective boundary layers on warm days, which may rise above 2 km, as observed in prior studies (e.g., Solanki et al., 2019). Thus, the MAA range (0.5–2.5 km) enables flexible detection without being constrained by the radiosonde's upper limit, while validation comparisons remain within the common 2 km vertical range. PBL heights were determined using the method of Wang and Wang (2014), which identifies the planetary boundary layer height by analyzing the first derivatives of key meteorological variables—specifically, temperature, wind speed, wind direction, potential temperature, dewpoint, and relative humidity. This approach considers both the maxima and minima in these gradients to detect significant atmospheric transitions associated with the top of the mixing layer. The final PBL height was computed as the average of the estimates derived from these parameters. This method was selected over traditional single-variable approaches because it integrates multiple physical parameters and accounts for cloud presence and stable stratification, providing more robust and consistent results under diverse atmospheric conditions. Earlier methods that rely solely on individual gradients (e.g., of potential temperature or humidity) are prone to inaccuracies, particularly in regions with residual layers, cloud-capped boundaries, or complex moisture profiles (Seibert et al., 2000; Liu and Liang, 2010). In contrast, the Wang and Wang method aligns discontinuities across multiple variables to better identify the true extent of turbulent mixing. Its integrative design makes it especially suitable for the complex atmospheric dynamics observed in this study over northern Thailand. However, a significant limitation is that the radiosondes were launched only once daily at 07:00 local time (00 UTC), coinciding with the early morning minimum PBL height. This constraint—stemming from the operational limitations of the Thai Meteorological Department—means that diurnal variations in the PBL height, especially during its daytime growth and decay phases, cannot be captured, potentially reducing the representativeness of radiosonde-derived estimates for broader atmospheric analyses. To address this limitation, future studies should incorporate higher temporal resolution datasets—such as those obtained from unmanned aerial vehicles (UAVs)—which can capture the full diurnal evolution of the boundary layer. Recent studies (e.g.,

Shen et al., 2023) have demonstrated the utility of UAV-based profiling for validating LiDAR-derived PBL heights under complex atmospheric conditions. Integrating UAV observations will strengthen validation and improve confidence in the performance of the proposed retrieval method across a wider range of temporal and meteorological regimes.

The simulations used in this study were run in forecast mode to reflect real operational conditions, with model output averaged over overlapping time periods in the forecast cycle to align with LiDAR data timestamps. Here, "overlapping time periods" refers specifically to the temporal matching between forecast outputs and observational sampling windows, and not to overlap correction in the LiDAR signal. WRF-Chem simulations were configured and optimized for mainland Southeast Asia (Bran et al., 2024) using version 4.3.3, with the Mellor-Yamada Nakanishi and Niino (MYNN) Level 3 PBL scheme (Olson et al., 2019). The simulations incorporated updated terrestrial data (Manomaiphiboon et al., 2017), anthropogenic emissions for northern Thailand (Jansakoo et al., 2019), and biogenic and fire emissions from MEGAN (Guenther et al., 2006) and FINNv1.5 (Wiedinmyer et al., 2011). To project fire emissions into the future (forecast mode), the assumption of persistent fire emissions was applied (Kumar et al., 2020). Initial and boundary conditions for meteorology and chemistry were derived from GFS (NCEP, 2024) and CESM2-WACCM (Gettelman et al., 2019), respectively. The horizontal spatial resolution used in the WRF-Chem simulation was 9-km covering mainland Southeast Asia. Since WRF-Chem uses a hybrid, sigma-pressure, terrain-following coordinate system, the vertical resolution used in this study with 38 vertical levels varies with altitude. Near the surface (0 to ~1,100 meters AGL; 24 vertical levels), it ranges from about 45–50 meters, increasing to approximately 70–250 meters in the lower troposphere (~1,100 to 2,000 meters AGL; 4 vertical levels). From the aerosol layer top to the mid troposphere (~2,000 to 7,000 meters AGL; 4 vertical levels), the resolution becomes coarser, ranging from 500 to 2,000 meters, and further coarsens to over 2,000 meters from the mid to the upper troposphere and stratosphere (~7,000 to 20,000 meters AGL; 6 vertical levels), with finer resolution near the surface to capture smaller-scale processes and coarser resolution at higher altitudes where larger-scale dynamics dominate. The WRF-Chem simulations used in this study have been previously validated under similar regional conditions in northern Thailand. Prior work has demonstrated the model's reliability in capturing key atmospheric dynamics, including surface $PM_{2.5}$ distributions, optical turbulence, and boundary layer processes. Notably, the model has been successfully applied in the following studies: Bran et al. (2022), Macatangay et al. (2024), and Bran et al. (2024). These validations support the robustness of WRF-Chem for use as a benchmark in our comparison with LiDAR-derived PBL heights.

This integrated methodology ensures better alignment of LiDAR-derived PBL heights with thermodynamic and aerosol boundaries, enhancing the reliability of boundary layer characterizations.

**3 Results and Discussion**

Figure 1 shows the NRB signal as a colored curtain plot, where the aerosol layer top (ALT) is marked as a white line, the time-varying maximum analysis altitude (MAA) as a gray line, and the refined PBL estimate as a red line. It is important to note

that the MAA does not represent the PBL height, but rather defines the maximum vertical range within which the wavelet covariance transform (WCT) analysis is conducted to detect the PBL height. For instance, if the MAA is set at 2.5 km AGL, the WCT algorithm only searches for the PBL height below that altitude. This adaptive constraint prevents overestimation of the PBL height, particularly during nighttime or during the growth and decay phases of the boundary layer. The MAA is defined dynamically and follows the diurnal variation of surface temperature, providing a physically realistic ceiling for analysis that adjusts with expected atmospheric mixing.

During 00:00–06:00 LT, conventional PBL detection methods often mischaracterize the PBL height by incorrectly identifying the residual layer top or aerosol layer top as the PBL. However, during the well-mixed part of the day and under cloud-free conditions (12:00–16:00 LT on January 27), the red PBL line closely aligns with the white ALT line. This alignment indicates a well-defined mixing layer, allowing for an accurate determination of the mixing layer height (MLH).

Under partly cloudy conditions (such as on January 28 during the afternoon between 12:00–16:00 LT), conventional algorithms misclassify the cloud base as the PBL height. In contrast, the refined red PBL line successfully separates from the white ALT line and follows the expected diurnal development, demonstrating improved reliability in characterizing the boundary layer, especially in aerosol-rich and meteorologically complex environments.

Transitional periods, such as the morning PBL growth phase (06:00–12:00) and evening decay (16:00–00:00), pose challenges due to aerosol accumulation in residual layers, which creates ambiguous gradients in the NRB signal. By incorporating the novel time-varying MAA and refining the PBL estimates, these limitations are mitigated.

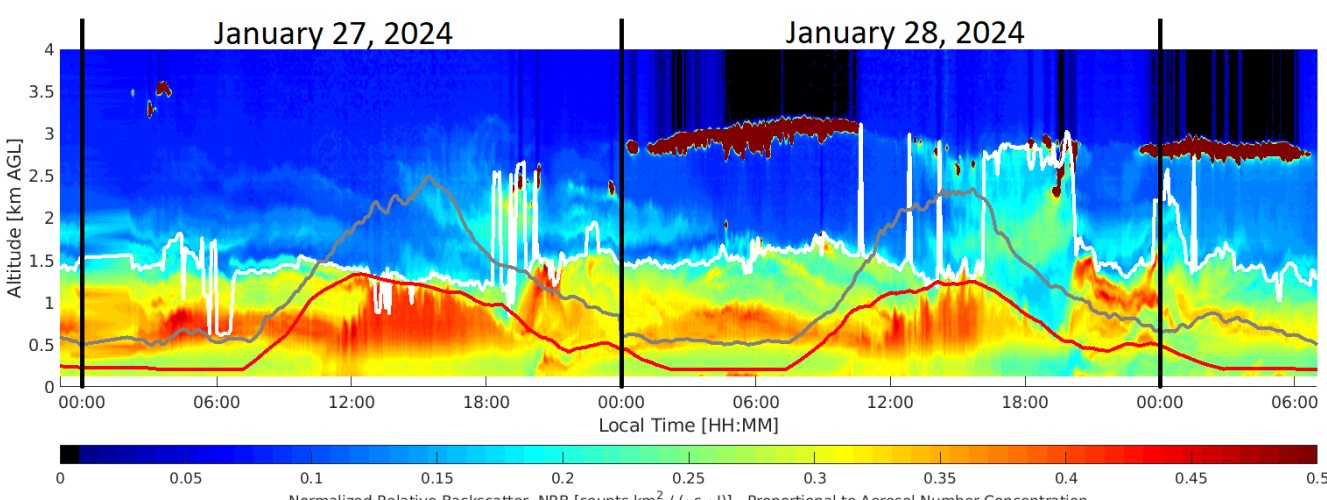

**Figure 1. The normalized relative backscatter (NRB) signal from the LiDAR is shown as a colored curtain plot, illustrating variations in aerosol number concentration over time. The aerosol layer top (ALT) is marked as a white line, the time-varying maximum analysis altitude (MAA) as a gray line, and the refined planetary boundary layer (PBL) estimate is shown in red. The MAA does not**

**represent the PBL height; rather, it defines the maximum altitude (in km AGL) up to which the WCT algorithm is applied to detect the PBL height. The MAA is dynamically adjusted based on surface temperature to follow the expected diurnal evolution of the boundary layer and to avoid overestimating the PBL height, particularly during nighttime and transitional phases. Data shown here were collected on January 27–29, 2024.**

Figures 2 and 3 demonstrate the improved alignment of these estimates with radiosonde data and highlight discrepancies with the WRF-Chem model simulations, respectively. Radiosonde-derived PBL heights, computed using first derivatives of meteorological parameters, provide a baseline comparison but are limited to once-daily measurements at 7 AM local time, often capturing the minimum PBL height. The temperature-based MAA method offers greater temporal resolution and adaptability, enhancing its reliability in capturing diurnal variations and transitions between the PBL and the ALT.

Figure 2 illustrates the variability of the PBL height and ALT under different atmospheric conditions for December 2023, January 2024, and February 2024. The LiDAR-derived PBL height (red curve) exhibits a pronounced diurnal cycle, peaking during the day due to solar-driven convection and decreasing at night under stable conditions. The radiosonde measurements (black points) closely align with the LiDAR-derived PBL heights, providing independent validation. However, their representativeness is limited by the once-daily launch at 7 AM local time (00 UTC), coinciding with the early morning minimum PBL height. In December 2023, the LiDAR-derived PBL height ranges between 0.1 km and 1.5 km AGL, with radiosonde values exhibiting strong agreement and a correlation coefficient of $r = 0.85$ and %RMSE of 10.0%. In January 2024, the LiDAR PBL height varied between 0.1 km and 1.6 km AGL, with excellent correlation to radiosonde measurements ($r = 0.88$; %RMSE = 7.2%). Similarly, in February 2024, the LiDAR-derived PBL height ranges from 0.1 km to 1.4 km AGL, maintaining a strong correlation with radiosonde data ($r = 0.94$; %RMSE = 19.6%), despite some divergence under complex atmospheric conditions. The ALT (green dashed line), representing the upper boundary of significant aerosol backscatter, often exceeds the PBL height during stable nighttime conditions, the presence of residual aerosol layers, or during aerosol entrainment into the free troposphere. By incorporating surface temperature variations, the proposed method effectively distinguishes the PBL height from the ALT, addressing challenges during transitions between daytime and nighttime conditions. While the PBL height is primarily governed by thermodynamic properties and turbulent mixing, the ALT reflects aerosol stratification and distribution. These layers typically align during well-mixed daytime conditions (mixing layer height), but they diverge under stratified or complex layering scenarios, emphasizing the necessity of distinguishing between them for accurate air quality modeling and pollutant transport analysis.

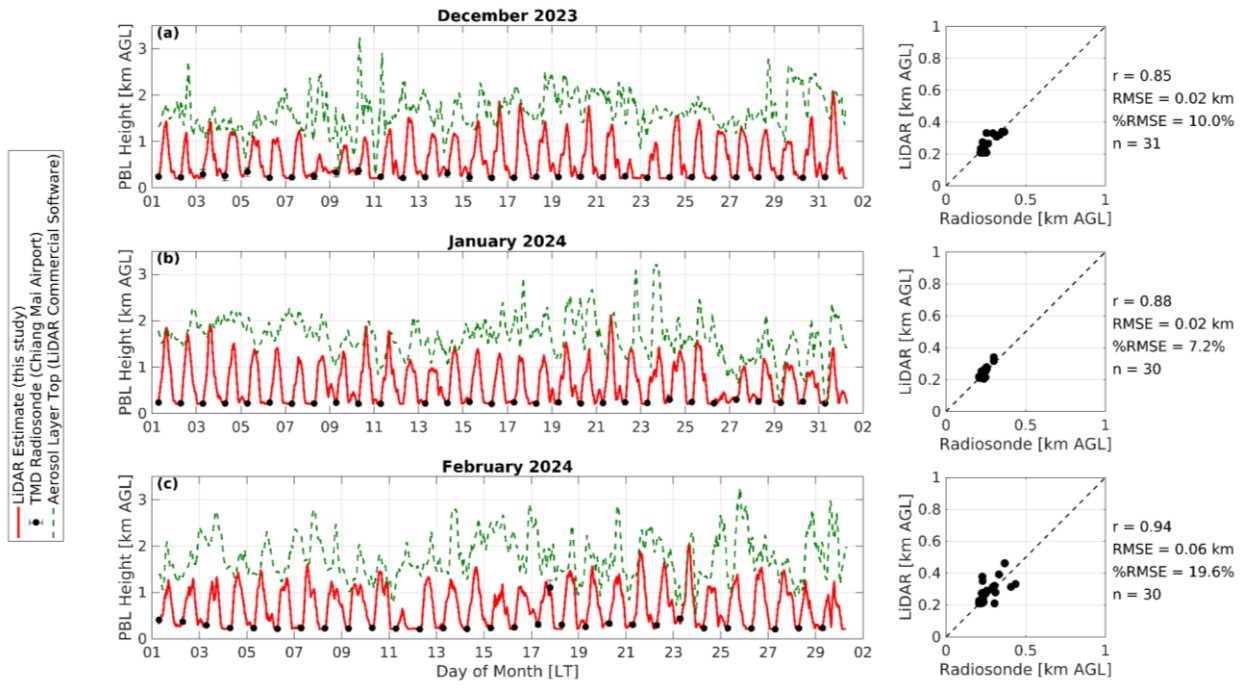

Figure 2. Comparison of planetary boundary layer (PBL) height estimates derived from this study's LiDAR retrievals (red curve), TMD radiosonde measurements at Chiang Mai Airport (black points), and aerosol layer top (ALT) heights calculated using commercial LiDAR software (green dashed line) for (a) December 2023, (b) January 2024, and (c) February 2024 at NARIT AstroPark, Chiang Mai, northern Thailand. The right panels show the correlations between LiDAR-based PBL estimates and radiosonde-derived heights, with Pearson correlation coefficients (r-values) and %RMSEs ranging from 0.85 to 0.94 and 7.2% to 19.6%, respectively, for each month.

Figure 3 compares PBL height estimates from LiDAR retrievals (red line) and WRF-Chem forecasts (black line) for December 2023, January 2024, and February 2024 at NARIT AstroPark in Chiang Mai, northern Thailand. While the WRF-Chem model generally captures seasonal and diurnal variations in PBL height, it tends to overestimate daytime PBL heights during periods of elevated aerosol loading. This behavior persisted across simulations using different PBL parameterizations (e.g., MYNN2.5, MYNN3, and YSU) and varied fire and anthropogenic emission inputs, suggesting a consistent model response to aerosol–meteorology coupling. The overestimation likely stems from limitations in how current model configurations, particularly those using the MYNN Level 3 scheme, represent aerosol–meteorology–radiation interactions and their feedbacks on turbulence generation and vertical mixing (Du et al., 2020; Petäjä et al., 2016). Correlation coefficients (r = 0.85, 0.86, and 0.81 for December, January, and February, respectively) indicate strong agreement between LiDAR retrievals and model outputs. However, the root mean square error (RMSE) and percentage RMSE (%RMSE) increased from December (0.28 km, 21%) to February (0.46 km, 33%), coinciding with peak fire activity. This degradation in model performance further supports the hypothesis that persistent fire emissions amplify the aerosol burden, modifying radiative transfer and boundary layer

stability in ways that are not fully captured by current parameterizations (Kumar et al., 2020). These findings underscore the need to refine PBL schemes and improve aerosol–radiation feedback representation in regional models operating under high aerosol loading conditions.

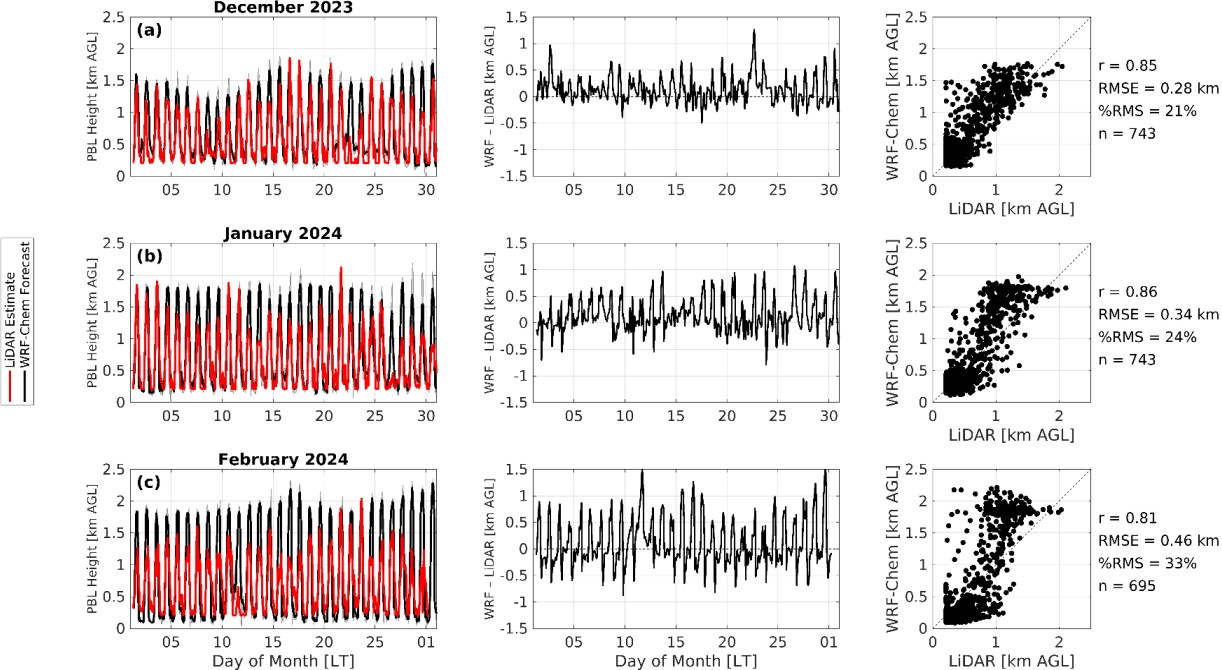

**Figure 3. Comparison of planetary boundary layer (PBL) height estimates from LiDAR retrievals (red line) and WRF-Chem forecasts (black line) at NARIT AstroPark, Chiang Mai, Thailand, for (a) December 2023, (b) January 2024, and (c) February 2024 (left column). The middle column shows time series of differences between WRF-Chem and LiDAR estimates (WRF – LiDAR), while the right column presents scatter plots with correlation coefficients (r), root mean square error (RMSE), percentage RMSE**
**(%RMSE), and number of matched data points (n). The WRF-Chem model configuration used in this comparison has been previously validated under similar regional conditions in northern Thailand, supporting its use as a reference for PBL height estimation.**

These results underscore the significance of dynamic adjustments in LiDAR analyses for improved boundary layer
characterization and the critical need to address limitations in both observational and modeling techniques for regions with complex atmospheric conditions.

## 4 Conclusions and Recommendations

This study successfully demonstrated the decoupling of the planetary boundary layer (PBL) height, mixing layer height (MLH), and aerosol layer top (ALT) using high-resolution LiDAR measurements in Chiang Mai, northern Thailand. By applying a novel temperature-based dynamic maximum analysis altitude (MAA) approach, the research overcame limitations of conventional methods, such as the Haar Wavelet Covariance Transform (WCT), which often misinterprets the ALT as the

PBL height, particularly during nighttime or transitional periods. The proposed method, integrating normalized relative
backscatter (NRB) profiles with surface temperature observations, improves the accuracy of PBL height estimations by accounting for variations in thermodynamic and atmospheric conditions. The results showed good agreement with both radiosonde data and WRF-Chem simulations (up to moderate aerosol loads), emphasizing the value of dynamic adjustments in refining boundary layer characterizations.

This study highlights the complexities of aerosol layering and PBL identification in regions with high aerosol loading, such as Chiang Mai, where seasonal forest fires and agricultural burning contribute to significant atmospheric pollution. By enhancing the accuracy of LiDAR-based PBL height estimations, the research provides critical insights for improving air quality modeling and understanding pollutant transport under complex meteorological conditions. The results demonstrate a generally consistent relationship between LiDAR-derived and model-based PBL height estimates, with seasonal variations in agreement
metrics. The WRF-Chem model configuration, previously validated over northern Thailand for air quality, turbulence, and PBL structure, continues to perform reliably as a comparison benchmark. These findings support the value of integrating high-resolution LiDAR retrievals with regional models to evaluate boundary layer processes and diagnose modeling uncertainties across different seasons.

Future research should focus on refining the dynamic MAA approach to LiDAR methodology for better handling of extreme atmospheric conditions, such as high aerosol loads and abrupt meteorological changes, while expanding its applicability to regions with diverse aerosol profiles to validate robustness. Integrating LiDAR with other remote sensing tools like radar or satellite-based sensors could enhance the accuracy and spatial resolution of PBL and aerosol layer measurements, particularly in areas with dense cloud cover or frequent atmospheric transitions. To address the limitations of once-daily radiosonde
measurements, more frequent launches or the adoption of continuous vertical profiling instruments is recommended to capture diurnal variations more effectively, providing a richer dataset for model validation. Comparisons of LiDAR-derived PBL estimates with simulations from models like WRF-Chem are essential to evaluate model accuracy under high-aerosol conditions and to improve parameterizations for aerosol transport and mixing. Finally, refined PBL height estimation methods can enhance air quality forecasting systems, improving pollutant dispersion predictions and supporting more effective public
health advisories and mitigation strategies in regions such as northern Thailand with significant seasonal aerosol emissions.

**Code Availability**

The code underlying this article will be shared on reasonable request to the corresponding author.

**Data Availability**

The data underlying this article will be shared on reasonable request to the corresponding author.

**Author Contribution**

R.M. conceived and led the study. He developed the novel PBL retrieval algorithm and the LiDAR–radiosonde–WRF-Chem comparison framework, designed the analysis pipeline, acquired funding, supervised all research phases, and was the main author of the original draft and manuscript revisions.

315 T.S. led the analysis of the LiDAR datasets and contributed to conceptual development in collaboration. She investigated the limitations of conventional planetary boundary layer (PBL) estimation based on the Haar wavelet covariance transform (WCT) technique, co-supervised field deployments, and co-refined the key retrieval algorithm, including statistical analysis and validation. She contributed to both the original draft and subsequent revisions, shaping the manuscript's scientific framing.

S.H.B. developed and implemented the WRF-Chem forecasting system and performed sensitivity analyses to optimize its configuration for the study region.

W.T. implemented the Haar wavelet covariance transform (WCT) based on the Ceilo code.

325 T.Su. secured project funding, coordinated with institutional stakeholders, and managed overall project logistics and reporting.

M.P. maintained the LiDAR system, ensuring continuous data acquisition throughout the campaign.

J.L. supported on-site logistics, including equipment transport and compliance with safety protocols.

R.S. contributed early insights on the limitations of conventional PBL estimation techniques and provided critical feedback during the manuscript editing phase.

B.S. indirectly contributed to conceptualization, methodology, and resource sharing through a parallel project on tethered 335 balloon profiling, which informed aspects of the LiDAR–radiosonde comparison.

A.H. indirectly supported the technical framework through his involvement in the tethered balloon project, contributing to methodological alignment and shared resources.

All authors reviewed and approved the final manuscript.

## Competing Interests

The authors declare that they have no competing interests.

## Acknowledgements

The authors would like to thank the following: Willsinee Thumrongchadthai, Panittha Chumpu, Kunakorn Kaewboonpan,
Aunnicha Kanipongthanasorn, Natthida Yarangsri, Thammarat Phaengmaphom, Jirasak Noisapung, Vichawan Sakulsupich,
Utane Sawangwit, Suchinno Kanthum, Supachai Awiphan, HPC, OPD and management teams of NARIT.

The authors would also like to acknowledge the use of AI tools to assist in improving the language and readability of the
manuscript. These tools were used solely to enhance clarity and ensure the manuscript is easily understood. No AI tools were
used to generate insights or analysis related to the research work presented.

## Financial Support

This study is funded by the Thailand Science Research and Innovation under the project no. FFB680072/0269.

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
