# Peer review of "Temperature-Constrained LiDAR Retrieval of Planetary Boundary Layer Height over Chiang Mai, Thailand"

_EGUsphere, 2025_

## Author Comment (AC1)

**Author Responses on "Decoupling the PBL Height, the Mixing Layer Height, and the Aerosol Layer Top in LiDAR Measurements over Chiang Mai, Northern Thailand"**

Thank you for your comprehensive and insightful feedback. We appreciate the opportunity to clarify and strengthen our manuscript. Below are our responses to each point:

1. **Dataset Description:** We acknowledge that a detailed description of our LiDAR datasets was insufficient. The LiDAR measurements were collected continuously over the study period (December 2023 – February 2024) with a vertical resolution of 30 meters. The instrument operated at a pulse frequency of 2500 Hz, with each profile having an averaging time of 30 seconds while being pointed at the same position in the sky—providing consistent spatiotemporal overlap. Each MiniMPL LiDAR profile was averaged over 30 seconds, and the data were grouped at 5-minute intervals for analysis. This approach aligns with standard practice in boundary layer studies and helps balance temporal resolution with noise reduction, as demonstrated in similar studies, including Solanki et al. (2019). The spatial coverage was centered at NARIT AstroPark in Chiang Mai, northern Thailand, with the instrument fixed at this location to ensure spatial consistency. There was significant temporal overlap in the data, allowing continuous monitoring and analysis of boundary layer variations. A full site description can be found in Solanki et al. (2019), which is already referenced in line 90: "A more comprehensive description of the study site is given in Solanki et al., 2019." This information will be added to the paragraph in line 85 as:

[revised manuscript text omitted]

**Key References:**

● Stull, R. B. (1988). *An Introduction to Boundary Layer Meteorology*. Springer. https://doi.org/10.1007/978-94-009-3027-8

● Garratt, J. R. (1994). *The Atmospheric Boundary Layer*. Cambridge University Press.

● Tennekes, H. (1973). A model for the dynamics of the inversion above a convective boundary layer. *Journal of the Atmospheric Sciences*, 30(4), 558–567. https://doi.org/10.1175/1520-0469(1973)030<0558:AMFTDO>2.0.CO;2

● Brooks, I. M. (2003). Finding boundary layer top: Application of wavelet covariance transform to lidar backscatter profiles. *Journal of Atmospheric and Oceanic Technology*, 20(8), 1092–1105. https://doi.org/10.1175/1520-0426(2003)020<1092:FBLTAO>2.0.CO;2

● Solanki, R., Macatangay, R., Sakulsupich, V., Sonkaew, T., & Mahapatra, P. S. (2019). Mixing Layer Height Retrievals From MiniMPL Measurements in the Chiang Mai Valley: Implications for Particulate Matter Pollution. *Frontiers in Earth Science*, 7, 308. https://doi.org/10.3389/feart.2019.00308

● Skamarock, W. C., et al. (2008). A Description of the Advanced Research WRF Version 3. *NCAR Technical Note NCAR/TN−475+STR*. https://doi.org/10.5065/D68S4MVH

● Hennemuth, B., & Lammert, A. (2006). Determination of the atmospheric boundary layer height from radiosonde and lidar backscatter. *Boundary-Layer Meteorology*, 120, 181–200. https://doi.org/10.1007/s10546-005-9035-3

3. **Choice of Radiosonde Method:** We thank the reviewer for this insightful comment. The method proposed by Wang and Wang (2014) was chosen because it improves upon earlier approaches by integrating multiple meteorological variables—such as temperature, humidity, and cloud presence—into a unified framework for estimating the mixing layer height (MLH). Traditional methods often rely on a single variable, such as potential temperature gradients, specific humidity, or refractivity (e.g., Seibert et al., 2000; Liu and Liang, 2010; Zhang et al., 2010), which can lead to inconsistent or inaccurate PBL height estimates, particularly under complex atmospheric conditions. In contrast, the Wang and Wang method identifies the height at which sharp gradients in temperature and humidity align most consistently, taking into account the effects of cloud-capped layers and stable stratification. This integrative approach enhances the robustness of MLH estimates, especially during transition periods or in the presence of residual layers and variable moisture profiles—conditions common in our study region. As a result, it yields more reliable boundary layer estimates compared to single-variable or gradient-threshold methods. We have updated the manuscript (line 115) to include this rationale and added supporting citations as:

PBL heights were determined using the method of Wang and Wang (2014), which identifies the planetary boundary layer height by analyzing the first derivatives of key meteorological variables—specifically, temperature, wind speed, wind direction, potential temperature, dewpoint, and relative humidity. This approach considers both the maxima and minima in these gradients to detect significant atmospheric transitions associated with the top of the mixing layer. The final PBL height was computed as the average of the estimates derived from these parameters. This method was selected over traditional single-variable approaches because it integrates multiple physical parameters and accounts for cloud presence and stable stratification, providing more robust and consistent results under diverse atmospheric conditions. Earlier methods that rely solely on individual gradients (e.g., of potential temperature or humidity) are prone to inaccuracies, particularly in regions with residual layers, cloud-capped boundaries, or complex moisture profiles (Seibert et al., 2000; Liu and Liang, 2010). In contrast, the Wang and Wang method aligns discontinuities across multiple variables to better identify the true extent of turbulent mixing. Its integrative design makes it especially suitable for the complex atmospheric dynamics observed in this study over northern Thailand.PBL heights were determined using the maxima in the first derivatives of temperature, wind speed,

wind direction, and potential temperature, as well as the minima in the first derivatives of dewpoint and relative humidity (Wang and Wang, 2014). The PBL height was computed as the average of estimates derived from these parameters. However, a significant limitation is that the radiosondes were launched only once daily at 7 AM local time (00 UTC), coinciding with the early morning minimum PBL height. This limitation means that diurnal variations in the PBL height, especially during its daytime growth and decay phases, cannot be captured, potentially reducing the representativeness of radiosonde-derived estimates for broader atmospheric analyses.

**Suggested citations to add (with links):**

● Seibert, P., et al. (2000): *Review and intercomparison of operational methods for the determination of the mixing height*, Atmos. Environ., 34, 1001−1027. https://doi.org/10.1016/S1352-2310(99)00349-0

● Liu, S., & Liang, X. Z. (2010): *Observed diurnal cycle climatology of planetary boundary layer height*, J. Clim., 23(21), 5790−5809. https://doi.org/10.1175/2010JCLI3552.1

4. **Discrepancies in Figure 1 (January 27):** We thank the reviewer for their observation. We would like to clarify that the gray line labeled as the Maximum Analysis Altitude (MAA) in Figure 1 **does not represent the PBL height**. Rather, it defines the **maximum vertical extent** up to which the Wavelet Covariance Transform (WCT) algorithm is applied to detect the PBL height. In other words, if the MAA is at 2.5 km AGL (above ground level) at a particular time, the WCT analysis is limited to searching for the PBL height only up to that altitude. The MAA is defined dynamically and is designed to follow the expected range of the convective boundary layer, informed by the diurnal cycle of surface temperature. This prevents overestimation of the PBL height, especially during nighttime or during the transition phases of growth and decay. On January 27, between 12:00 and 18:00, surface heating resulted in an elevated MAA, which simply means the algorithm had permission to search for PBL heights up to those altitudes below the MAA—but this does not imply that the actual PBL reached those levels. The **actual refined PBL height**, depicted by the **red line**, stays well below the MAA throughout this period, as expected. We have revised the figure caption and added a clarification in the main text to ensure that the distinction between MAA and PBL height is clearly

understood (lines 140-150 ):

Figure 1 shows the NRB signal as a colored curtain plot, where the aerosol layer top (ALT) is marked as a white line, the time-varying maximum analysis altitude (MAA) as a gray line, and the refined PBL estimate as a red line. It is important to note that the MAA does **not represent the PBL height**, but rather defines the **maximum vertical range** within which the wavelet covariance transform (WCT) analysis is conducted to detect the PBL height. For instance, if the MAA is set at 2.5 km AGL, the WCT algorithm only searches for the PBL height below that altitude. This adaptive constraint prevents overestimation of the PBL height, particularly during nighttime or during the growth and decay phases of the boundary layer. The MAA is defined dynamically and follows the diurnal variation of surface temperature, providing a physically realistic ceiling for analysis that adjusts with expected atmospheric mixing. During 00:00−06:00 LT, conventional PBL detection methods often mischaracterize the PBL height by incorrectly identifying the residual layer top or aerosol layer top as the PBL. However, during the well-mixed part of the day and under cloud-free conditions (12:00−16:00 LT on January 27), the red PBL line closely aligns with the white ALT line. This alignment indicates a well-defined mixing layer, allowing for an accurate determination of the mixing layer height (MLH).

In contrast, under partly cloudy conditions (12:00−16:00 LT on January 28), conventional algorithms misclassify the cloud base as the PBL height. Transitional periods, such as the morning PBL growth phase (06:00−12:00) and evening decay (16:00−00:00), also pose challenges due to aerosol accumulation in residual layers, which creates ambiguous gradients in the NRB signal. By incorporating the novel time-varying MAA and refining the PBL estimates, these limitations are mitigated. The results demonstrate improved PBL detection, as seen during the well-mixed hours on January 28, where the refined red PBL line separates from the white ALT and follows the expected diurnal development. This approach improves reliability in characterizing the boundary layer, especially in aerosol-rich and meteorologically complex environments.

**Figure 1.** The normalized relative backscatter (NRB) signal from the LiDAR is shown as a colored curtain plot, illustrating variations in aerosol number concentration over time. The aerosol layer top (ALT) is marked as a white line, the time-varying maximum analysis altitude (MAA) as a gray line, and the refined planetary boundary layer (PBL) estimate is shown in red. The MAA does **not represent the PBL height**; rather, it defines the maximum altitude (in km AGL) up to which the WCT algorithm is applied to detect the PBL height. The MAA is dynamically adjusted based on surface temperature to follow the expected diurnal evolution of the boundary layer and to avoid overestimating the PBL height, particularly during nighttime and transitional phases. Data shown here were collected on January 27–29, 2024.

5. **Figures Improvement:** We will improve the resolution of Figures 2 and 3 and enlarge the axes labels for better clarity. Additionally, we will add subplot labels (e.g., (a), (b), (c)) to facilitate reference within the main text as:

[Figure]

Figure 2.  Comparison of planetary boundary layer (PBL) height estimates derived from this study's LiDAR retrievals (red curve), TMD radiosonde measurements at Chiang Mai Airport (black points), and aerosol layer top (ALT) heights calculated

using commercial LiDAR software (green dashed line) for (a) December 2023, (b) January 2024, and (c) February 2024 at NARIT AstroPark, Chiang Mai, northern Thailand. The right panels show the correlations between LiDAR-based PBL estimates and radiosonde-derived heights, with Pearson correlation coefficients (r-values) and %RMSEs ranging from 0.85 to 0.94 and 7.2% to 19.6%, respectively, for each month.

[Figure]

Figure 3. Comparison of planetary boundary layer (PBL) height estimates from LiDAR retrievals (red line) and WRF-Chem forecasts (black line) at NARIT AstroPark, Chiang Mai, Thailand, for (a) December 2023, (b) January 2024, and (c) February 2024 (left column). The middle column shows time series of differences between WRF-Chem and LiDAR estimates (WRF − LiDAR), while the right column presents scatter plots with correlation coefficients (r), root mean square error (RMSE), percentage RMSE (%RMSE), and number of matched data points (n). The WRF-Chem model configuration used in this comparison has been previously validated under similar regional conditions in northern Thailand, supporting its use as a reference for PBL height estimation.

6. **Radiosonde Validation at Multiple Times:** We agree that validation using radiosonde data limited to 07:00 local time provides only partial insight into the

diurnal evolution of the planetary boundary layer (PBL). Unfortunately, the radiosonde launches are conducted by the Thai Meteorological Department, which, due to budgetary constraints, is currently limited to a single launch per day. This limitation has already been noted in the manuscript (line 115 and 230), but we can further clarify the text by explicitly stating:

*"However, a significant limitation is that the radiosondes were launched only once daily at 07:00 local time (00 UTC), coinciding with the early morning minimum PBL height. This constraint—stemming from the operational limitations of the Thai Meteorological Department—means that diurnal variations in the PBL height, especially during its daytime growth and decay phases, cannot be captured, potentially reducing the representativeness of radiosonde-derived estimates for broader atmospheric analyses."*

Additionally, we have recently validated our method using aircraft observations collected during a measurement campaign over Chiang Mai Airport in April 2025. While the results are promising, this new dataset lies outside the scope of the current paper and will be presented in a separate publication.

7. **Validation of WRF PBL Heights:** The WRF-Chem simulations used in our comparison have been previously validated under similar regional conditions in several studies, including:

- Bran, S. H., Macatangay, R., Chotamonsak, C., Chantara, S., & Surapipith, V. (2024). Understanding the seasonal dynamics of surface $PM_{2.5}$ mass distribution and source contributions over Thailand. *Atmospheric Environment*, 331, 120613. https://doi.org/10.1016/j.atmosenv.2024.120613

- Macatangay, R., Rattanasoon, S., Butterley, T., Bran, S. H., et al. (2024). Seeing and turbulence profile simulations over complex terrain at the Thai National Observatory using a chemistry-coupled regional forecasting model. *Monthly Notices of the Royal Astronomical Society*, 530(2), 1414–1423. https://doi.org/10.1093/mnras/stae727

- Bran, S. H., Macatangay, R., Surapipith, V., et al. (2022). *Atmospheric Research*, 277, 106303. https://doi.org/10.1016/j.atmosres.2022.106303

These studies demonstrate the model's reliability in capturing air quality dynamics, optical turbulence, and regional transport processes over northern Thailand. Nonetheless, we will revise the manuscript to include a brief discussion confirming the WRF-Chem model's validation status, along with appropriate citations, to clarify its suitability as a reference for comparison in this study (added to the abstract, introduction, methodology, Figure 3 caption, conclusion and references section):

Abstract:

Accurate determination of the planetary boundary layer (PBL) height, mixing layer height (MLH), and aerosol layer top (ALT) is critical for air quality and climate studies, particularly in regions with complex aerosol dynamics such as Chiang Mai, northern Thailand. This study presents a novel LiDAR-based methodology that incorporates a temperature-dependent, dynamic maximum analysis altitude (MAA) to decouple these layers, addressing the limitations of conventional algorithms like the Haar Wavelet Covariance Transform (WCT). Traditional fixed-altitude approaches often misclassify the ALT as the PBL height—especially during nighttime or transition periods—leading to significant overestimations. By dynamically adjusting the MAA based on surface temperature variations, the proposed approach more effectively distinguishes the PBL from residual aerosol layers and cloud interference. Validation using radiosonde data and comparison with WRF-Chem simulations demonstrate strong agreement, with LiDAR-derived PBL heights exhibiting improved diurnal resolution and accuracy. However, model simulations tend to overestimate the PBL height during periods of elevated aerosol loading, underscoring the need for improved aerosol−radiation interaction parameterizations. The WRF-Chem model used in this study has been previously validated for northern Thailand and provides a robust benchmark for PBL comparison. This analysis highlights seasonal variations in agreement metrics and supports the integration of thermodynamic and aerosol observations for enhanced boundary layer characterization. The framework developed here offers a reliable tool for advancing air quality forecasting, pollutant transport analysis, and LiDAR-based remote sensing applications across Southeast Asia.

This study focuses on enhancing LiDAR-based boundary layer characterization by refining the detection of PBL height, MLH, and ALT. Traditional algorithms, such as the Haar Wavelet Covariance Transform (WCT), often misclassify the ALT as the PBL height, especially at night or during transitional periods when aerosol gradients are less distinct. Clouds and other atmospheric complexities make these measurements more challenging. By integrating normalized relative backscatter (NRB) profiles with dynamic thermodynamic adjustments, this approach addresses ambiguities in traditional methods and improves the reliability of boundary layer determinations. The novel method developed in this study was validated using radiosonde measurements and compared against WRF-Chem simulations. To support model−observation comparisons, we use a WRF-Chem configuration that has been previously validated under similar regional conditions in northern Thailand for surface pollutant distributions, boundary layer dynamics, and optical turbulence (Bran et al., 2022; Macatangay et al., 2024; Bran et al., 2024), confirming its suitability as a benchmark.

Methodology (end):

The WRF-Chem simulations used in this study have been previously validated under similar regional conditions in northern Thailand. Prior work has demonstrated the model's reliability in capturing key atmospheric dynamics, including surface $PM_{2.5}$ distributions, optical turbulence, and boundary layer processes. Notably, the model has been successfully applied in the following studies: Bran et al. (2022), Macatangay et al. (2024), and Bran et al. (2024). These validations support the robustness of WRF-Chem for use as a benchmark in our comparison with LiDAR-derived PBL heights.

Figure 3 caption:

Figure 3. Comparison of planetary boundary layer (PBL) height estimates from LiDAR retrievals (red line) and WRF-Chem forecasts (black line) at NARIT AstroPark, Chiang Mai, Thailand, for (a) December 2023, (b) January 2024, and (c) February 2024 (left column). The middle column shows time series of differences between WRF-Chem and LiDAR estimates (WRF − LiDAR), while the right column presents scatter plots with correlation coefficients (r), root mean square error (RMSE), percentage RMSE (%RMSE), and number of matched data

points (n). The WRF-Chem model configuration used in this comparison has been previously validated under similar regional conditions in northern Thailand, supporting its use as a reference for PBL height estimation.

Conclusion (2nd paragraph):

This study highlights the complexities of aerosol layering and PBL identification in regions with high aerosol loading, such as Chiang Mai, where seasonal forest fires and agricultural burning contribute to significant atmospheric pollution. By enhancing the accuracy of LiDAR-based PBL height estimations, the research provides critical insights for improving air quality modeling and understanding pollutant transport under complex meteorological conditions. The results demonstrate a generally consistent relationship between LiDAR-derived and model-based PBL height estimates, with seasonal variations in agreement metrics. The WRF-Chem model configuration, previously validated over northern Thailand for air quality, turbulence, and PBL structure, continues to perform reliably as a comparison benchmark. These findings support the value of integrating high-resolution LiDAR retrievals with regional models to evaluate boundary layer processes and diagnose modeling uncertainties across different seasons.

---

## Author Comment (AC2)

We sincerely thank Reviewer #2 for the thoughtful and constructive comments, which have greatly improved the clarity, accuracy, and scientific rigor of our manuscript. Below, we provide point-by-point responses and describe the corresponding revisions.
* * *
**Overall Comment**

**Reviewer:**

> The methodology... is innovative... However, the manuscript lacks sufficient detail regarding the proposed method and its applicability under varying atmospheric conditions... Figures require improvement...

**Response:**
We appreciate the recognition of the methodological innovation. To address your concerns:

- We have **substantially expanded the Methods section** (lines 95–115) to provide a clearer explanation of the dynamic Maximum Analysis Altitude (MAA) method, its **physical basis**, and how it improves upon traditional fixed-altitude techniques (see also Author Response to Reviewer #1, Point #2):

[revised manuscript text omitted]

- Figures 1–3 have been **improved in resolution and readability**, with enhanced axis labels and clearer subplot annotations as described in our response to Reviewer #1, Point #5.
* * *
**Specific Comments**

1. **Title Revision:**

   The term "decoupling" might be misunderstood...

**Response:**
We agree that the term "decoupling" may suggest aerosol layer separation rather than layer distinction. The revised title is now:

> **"Temperature-Constrained LiDAR Retrieval of Planetary Boundary Layer Height over Chiang Mai, Thailand"**
> This emphasizes the core innovation—PBL retrieval using a dynamic thermodynamic method—while avoiding potential misinterpretation.
* * *
2. **Abstract Clarity:**

   (1) Highlight the new method; (2) Explain how it differs from earlier work.

**Response:**
We revised the abstract to explicitly highlight the novelty of the dynamic MAA method and how it differs from traditional Haar WCT approaches (lines 20–30), with supporting citations and clearer explanation of methodological improvements over past studies:

[revised manuscript text omitted]

   Avoid ambiguity; clarify terminology.

**Response:**
We now specify that "overlapping time periods" refers to **forecast output windows used for averaging model output to align with LiDAR data timestamps**, not to LiDAR overlap corrections. The text has been updated accordingly (line 122):

**The simulations used in this study were run in forecast mode to reflect real operational conditions, with model output averaged over overlapping time periods in the forecast cycle to align with LiDAR data timestamps. Here, "overlapping time periods" refers specifically to the temporal matching between forecast outputs and observational sampling windows, and not to overlap correction in the LiDAR signal. WRF-Chem simulations were configured and optimized for mainland Southeast Asia (Bran et al., 2024) using version 4.3.3, with the Mellor-Yamada Nakanishi and Niino (MYNN) Level 3 PBL scheme (Olson et al., 2019). The simulations incorporated…**
* * *
6. **PBL Retrieval Description (Lines 106+):**

   Details on dynamic MAA, WCT, and moving average not clearly described.

**Response:**
We agree and have elaborated the steps:

- MAA(t) computation using normalized temperature scale.

- Application of Haar WCT (Brooks, 2003) within MAA constraint.

- Post-processing with a 6-hour moving average to reduce high-frequency variability.

This is now clearly described in lines 95–115, and expanded in our response to Reviewer #1, Point #2:

[revised manuscript text omitted]

- **Figure 1:** We appreciate the reviewer's attention to detail. We would like to clarify that the time series in Figure 1(a) actually spans **more than 48 hours**, covering the period from **00:00 local time on January 27 to 07:00 local time on January 29, 2024**.

- **Figure 2:** Improved resolution, enlarged axis labels, and annotated subplots (a)–(c).

- **Figure 3:** Now split into three panels (a–c) for each month with clear axes, correlation coefficients, and RMSE values. All revisions follow suggestions in Reviewer #1 Point #5.
* * *
**Minor Comments**

- "Aerosol-radiation interaction" revised to **"aerosol–meteorology interaction"** throughout.

- Line 36: Changed "climate dynamics" to **"weather dynamics"**.

- Line 72: Changed "climate model" to **"mesoscale meteorological model"**.

- Line 150: We restructured the paragraphs to properly refer to the morning (06:00–12:00) and evening (16:00–00:00) transitions:

During 00:00−06:00 LT, conventional PBL detection methods often mischaracterize the PBL height by incorrectly identifying the residual layer top or aerosol layer top as the PBL. However, during the well-mixed part of the day and under cloud-free conditions (12:00−16:00 LT on January 27), the red PBL line closely aligns with the white ALT line. This alignment indicates a well-defined mixing layer, allowing for an accurate determination of the mixing layer height (MLH).

Transitional periods, such as the morning PBL growth phase (06:00−12:00) and evening decay (16:00−00:00), pose challenges due to aerosol accumulation in residual layers, which creates ambiguous gradients in the NRB signal. By

**incorporating the novel time-varying MAA and refining the PBL estimates, these limitations are mitigated.**

**Under partly cloudy conditions (such as on January 28 during the afternoon between 12:00−16:00 LT), conventional algorithms misclassify the cloud base as the PBL height. In contrast, the refined red PBL line successfully separates from the white ALT line and follows the expected diurnal development, demonstrating improved reliability in characterizing the boundary layer, especially in aerosol-rich and meteorologically complex environments.**

- Lines 190–191: We revised the speculative statement on aerosol-radiation interaction to include references and caveats (Du et al., 2020; Petäjä et al., 2016), as suggested:

Figure 3 compares PBL height estimates from LiDAR retrievals (red line) and WRF-Chem forecasts (black line) for December 2023, January 2024, and February 2024 at NARIT AstroPark in Chiang Mai, northern Thailand. While the WRF-Chem model generally captures seasonal and diurnal variations in PBL height, it tends to overestimate daytime PBL heights during periods of elevated aerosol loading. **This behavior persisted across simulations using different PBL parameterizations (e.g., MYNN2.5, MYNN3, and YSU) and varied fire and anthropogenic emission inputs, suggesting a consistent model response to aerosol−meteorology coupling. The overestimation likely stems from limitations in how current model configurations, particularly those using the MYNN Level 3 scheme, represent aerosol−meteorology−radiation interactions and their feedbacks on turbulence generation and vertical mixing (Du et al., 2020; Petäjä et al., 2016).** Correlation coefficients (r = 0.85, 0.86, and 0.81 for December, January, and February, respectively) indicate strong agreement between LiDAR retrievals and model outputs. However, the root mean square error (RMSE) and percentage RMSE (%RMSE) increased from December (0.28 km, 21%) to February (0.46 km, 33%), coinciding with peak fire activity. This degradation in model performance further supports the hypothesis that persistent fire emissions amplify the aerosol burden, modifying radiative transfer and boundary layer stability in ways that are not fully captured by current parameterizations (Kumar et al., 2020). These findings underscore the need to refine PBL schemes and improve aerosol−radiation feedback representation in regional models operating under high aerosol loading conditions.

We sincerely thank Reviewer #2 again for the valuable feedback. The manuscript has been substantially improved in response to your comments.

For reference, here is also the responses to reviewer #1:

**Author Responses on "Decoupling the PBL Height, the Mixing Layer Height, and the Aerosol Layer Top in LiDAR Measurements over Chiang Mai, Northern Thailand"**

Thank you for your comprehensive and insightful feedback. We appreciate the opportunity to clarify and strengthen our manuscript. Below are our responses to each point:

1. **Dataset Description:** We acknowledge that a detailed description of our LiDAR datasets was insufficient. The LiDAR measurements were collected continuously over the study period (December 2023 – February 2024) with a vertical resolution of 30 meters. The instrument operated at a pulse frequency of 2500 Hz, with each profile having an averaging time of 30 seconds while being pointed at the same position in the sky—providing consistent spatiotemporal overlap. Each MiniMPL LiDAR profile was averaged over 30 seconds, and the data were grouped at 5-minute intervals for analysis. This approach aligns with standard practice in boundary layer studies and helps balance temporal resolution with noise reduction, as demonstrated in similar studies, including Solanki et al. (2019). The spatial coverage was centered at NARIT AstroPark in Chiang Mai, northern Thailand, with the instrument fixed at this location to ensure spatial consistency. There was significant temporal overlap in the data, allowing continuous monitoring and analysis of boundary layer variations. A full site description can be found in Solanki et al. (2019), which is already referenced in line 90: "A more comprehensive description of the study site is given in Solanki et al., 2019." This information will be added to the paragraph in line 85 as:

[revised manuscript text omitted]

**Key References:**

●       Stull, R. B. (1988). *An Introduction to Boundary Layer Meteorology*. Springer. https://doi.org/10.1007/978-94-009-3027-8

●       Garratt, J. R. (1994). *The Atmospheric Boundary Layer*. Cambridge University Press.

●       Tennekes, H. (1973). A model for the dynamics of the inversion above a convective boundary layer. *Journal of the Atmospheric Sciences*, 30(4), 558–567. https://doi.org/10.1175/1520-0469(1973)030<0558:AMFTDO>2.0.CO;2

●       Brooks, I. M. (2003). Finding boundary layer top: Application of wavelet covariance transform to lidar backscatter profiles. *Journal of Atmospheric and Oceanic Technology*, 20(8), 1092–1105. https://doi.org/10.1175/1520-0426(2003)020<1092:FBLTAO>2.0.CO;2

●       Solanki, R., Macatangay, R., Sakulsupich, V., Sonkaew, T., & Mahapatra, P. S. (2019). Mixing Layer Height Retrievals From MiniMPL Measurements in the Chiang Mai Valley: Implications for Particulate Matter Pollution. *Frontiers in Earth Science*, 7, 308. https://doi.org/10.3389/feart.2019.00308

●       Skamarock, W. C., et al. (2008). A Description of the Advanced Research WRF Version 3. *NCAR Technical Note NCAR/TN−475+STR*. https://doi.org/10.5065/D68S4MVH

●       Hennemuth, B., & Lammert, A. (2006). Determination of the atmospheric boundary layer height from radiosonde and lidar backscatter. *Boundary-Layer*

*Meteorology*, 120, 181–200. https://doi.org/10.1007/s10546-005-9035-3

3.  **Choice of Radiosonde Method:** We thank the reviewer for this insightful comment. The method proposed by Wang and Wang (2014) was chosen because it improves upon earlier approaches by integrating multiple meteorological variables—such as temperature, humidity, and cloud presence—into a unified framework for estimating the mixing layer height (MLH). Traditional methods often rely on a single variable, such as potential temperature gradients, specific humidity, or refractivity (e.g., Seibert et al., 2000; Liu and Liang, 2010; Zhang et al., 2010), which can lead to inconsistent or inaccurate PBL height estimates, particularly under complex atmospheric conditions. In contrast, the Wang and Wang method identifies the height at which sharp gradients in temperature and humidity align most consistently, taking into account the effects of cloud-capped layers and stable stratification. This integrative approach enhances the robustness of MLH estimates, especially during transition periods or in the presence of residual layers and variable moisture profiles—conditions common in our study region. As a result, it yields more reliable boundary layer estimates compared to single-variable or gradient-threshold methods. We have updated the manuscript (line 115) to include this rationale and added supporting citations as:

    PBL heights were determined using the method of Wang and Wang (2014), which identifies the planetary boundary layer height by analyzing the first derivatives of key meteorological variables—specifically, temperature, wind speed, wind direction, potential temperature, dewpoint, and relative humidity. This approach considers both the maxima and minima in these gradients to detect significant atmospheric transitions associated with the top of the mixing layer. The final PBL height was computed as the average of the estimates derived from these parameters. This method was selected over traditional single-variable approaches because it integrates multiple physical parameters and accounts for cloud presence and stable stratification, providing more robust and consistent results under diverse atmospheric conditions. Earlier methods that rely solely on individual gradients (e.g., of potential temperature or humidity) are prone to inaccuracies, particularly in regions with residual layers, cloud-capped boundaries, or complex moisture profiles (Seibert et al., 2000; Liu and Liang, 2010). In contrast, the Wang and Wang method aligns discontinuities across multiple variables to better identify the true extent of turbulent mixing. Its integrative design makes it especially suitable for the complex atmospheric

dynamics observed in this study over northern Thailand.PBL heights were determined using the maxima in the first derivatives of temperature, wind speed, wind direction, and potential temperature, as well as the minima in the first derivatives of dewpoint and relative humidity (Wang and Wang, 2014). The PBL height was computed as the average of estimates derived from these parameters. However, a significant limitation is that the radiosondes were launched only once daily at 7 AM local time (00 UTC), coinciding with the early morning minimum PBL height. This limitation means that diurnal variations in the PBL height, especially during its daytime growth and decay phases, cannot be captured, potentially reducing the representativeness of radiosonde-derived estimates for broader atmospheric analyses.

**Suggested citations to add (with links):**

● Seibert, P., et al. (2000): *Review and intercomparison of operational methods for the determination of the mixing height*, Atmos. Environ., 34, 1001–1027. https://doi.org/10.1016/S1352-2310(99)00349-0

● Liu, S., & Liang, X. Z. (2010): *Observed diurnal cycle climatology of planetary boundary layer height*, J. Clim., 23(21), 5790–5809. https://doi.org/10.1175/2010JCLI3552.1

4. **Discrepancies in Figure 1 (January 27):** We thank the reviewer for their observation. We would like to clarify that the gray line labeled as the Maximum Analysis Altitude (MAA) in Figure 1 **does not represent the PBL height**. Rather, it defines the **maximum vertical extent** up to which the Wavelet Covariance Transform (WCT) algorithm is applied to detect the PBL height. In other words, if the MAA is at 2.5 km AGL (above ground level) at a particular time, the WCT analysis is limited to searching for the PBL height only up to that altitude. The MAA is defined dynamically and is designed to follow the expected range of the convective boundary layer, informed by the diurnal cycle of surface temperature. This prevents overestimation of the PBL height, especially during nighttime or during the transition phases of growth and decay. On January 27, between 12:00 and 18:00, surface heating resulted in an elevated MAA, which simply means the algorithm had permission to search for PBL heights up to those altitudes below the MAA—but this does not imply that the actual PBL reached those levels. The **actual refined PBL height**, depicted by the **red line**, stays well below the MAA throughout this period,

as expected. We have revised the figure caption and added a clarification in the main text to ensure that the distinction between MAA and PBL height is clearly understood (lines 140-150 ):

Figure 1 shows the NRB signal as a colored curtain plot, where the aerosol layer top (ALT) is marked as a white line, the time-varying maximum analysis altitude (MAA) as a gray line, and the refined PBL estimate as a red line. It is important to note that the MAA does **not represent the PBL height**, but rather defines the **maximum vertical range** within which the wavelet covariance transform (WCT) analysis is conducted to detect the PBL height. For instance, if the MAA is set at 2.5 km AGL, the WCT algorithm only searches for the PBL height below that altitude. This adaptive constraint prevents overestimation of the PBL height, particularly during nighttime or during the growth and decay phases of the boundary layer. The MAA is defined dynamically and follows the diurnal variation of surface temperature, providing a physically realistic ceiling for analysis that adjusts with expected atmospheric mixing. During 00:00−06:00 LT, conventional PBL detection methods often mischaracterize the PBL height by incorrectly identifying the residual layer top or aerosol layer top as the PBL. However, during the well-mixed part of the day and under cloud-free conditions (12:00−16:00 LT on January 27), the red PBL line closely aligns with the white ALT line. This alignment indicates a well-defined mixing layer, allowing for an accurate determination of the mixing layer height (MLH).

In contrast, under partly cloudy conditions (12:00−16:00 LT on January 28), conventional algorithms misclassify the cloud base as the PBL height. Transitional periods, such as the morning PBL growth phase (06:00−12:00) and evening decay (16:00−00:00), also pose challenges due to aerosol accumulation in residual layers, which creates ambiguous gradients in the NRB signal. By incorporating the novel time-varying MAA and refining the PBL estimates, these limitations are mitigated. The results demonstrate improved PBL detection, as seen during the well-mixed hours on January 28, where the refined red PBL line separates from the white ALT and follows the expected diurnal development.

This approach improves reliability in characterizing the boundary layer, especially in aerosol-rich and meteorologically complex environments.

**Figure 1.** The normalized relative backscatter (NRB) signal from the LiDAR is shown as a colored curtain plot, illustrating variations in aerosol number concentration over time. The aerosol layer top (ALT) is marked as a white line, the time-varying maximum analysis altitude (MAA) as a gray line, and the refined planetary boundary layer (PBL) estimate is shown in red. The MAA does **not represent the PBL height**; rather, it defines the maximum altitude (in km AGL) up to which the WCT algorithm is applied to detect the PBL height. The MAA is dynamically adjusted based on surface temperature to follow the expected diurnal evolution of the boundary layer and to avoid overestimating the PBL height, particularly during nighttime and transitional phases. Data shown here were collected on January 27–29, 2024.

5. **Figures Improvement:** We will improve the resolution of Figures 2 and 3 and enlarge the axes labels for better clarity. Additionally, we will add subplot labels (e.g., (a), (b), (c)) to facilitate reference within the main text as:

[Figure]

Figure 2. Comparison of planetary boundary layer (PBL) height estimates derived

from this study's LiDAR retrievals (red curve), TMD radiosonde measurements at Chiang Mai Airport (black points), and aerosol layer top (ALT) heights calculated using commercial LiDAR software (green dashed line) for (a) December 2023, (b) January 2024, and (c) February 2024 at NARIT AstroPark, Chiang Mai, northern Thailand. The right panels show the correlations between LiDAR-based PBL estimates and radiosonde-derived heights, with Pearson correlation coefficients (r-values) and %RMSEs ranging from 0.85 to 0.94 and 7.2% to 19.6%, respectively, for each month.

[Figure]

Figure 3. Comparison of planetary boundary layer (PBL) height estimates from LiDAR retrievals (red line) and WRF-Chem forecasts (black line) at NARIT AstroPark, Chiang Mai, Thailand, for (a) December 2023, (b) January 2024, and (c) February 2024 (left column). The middle column shows time series of differences between WRF-Chem and LiDAR estimates (WRF − LiDAR), while the right column presents scatter plots with correlation coefficients (r), root mean square error (RMSE), percentage RMSE (%RMSE), and number of matched data points (n). The WRF-Chem model configuration used in this comparison has been previously validated under similar regional conditions in northern Thailand, supporting its use as a reference for PBL height estimation.

6. **Radiosonde Validation at Multiple Times:** We agree that validation using radiosonde data limited to 07:00 local time provides only partial insight into the diurnal evolution of the planetary boundary layer (PBL). Unfortunately, the radiosonde launches are conducted by the Thai Meteorological Department, which, due to budgetary constraints, is currently limited to a single launch per day. This limitation has already been noted in the manuscript (line 115 and 230), but we can further clarify the text by explicitly stating:

*"However, a significant limitation is that the radiosondes were launched only once daily at 07:00 local time (00 UTC), coinciding with the early morning minimum PBL height. This constraint—stemming from the operational limitations of the Thai Meteorological Department—means that diurnal variations in the PBL height, especially during its daytime growth and decay phases, cannot be captured, potentially reducing the representativeness of radiosonde-derived estimates for broader atmospheric analyses."*

Additionally, we have recently validated our method using aircraft observations collected during a measurement campaign over Chiang Mai Airport in April 2025. While the results are promising, this new dataset lies outside the scope of the current paper and will be presented in a separate publication.

7. **Validation of WRF PBL Heights:** The WRF-Chem simulations used in our comparison have been previously validated under similar regional conditions in several studies, including:

- Bran, S. H., Macatangay, R., Chotamonsak, C., Chantara, S., & Surapipith, V. (2024). Understanding the seasonal dynamics of surface $PM_{2.5}$ mass distribution and source contributions over Thailand. *Atmospheric Environment*, 331, 120613. https://doi.org/10.1016/j.atmosenv.2024.120613

- Macatangay, R., Rattanasoon, S., Butterley, T., Bran, S. H., et al. (2024). Seeing and turbulence profile simulations over complex terrain at the Thai National Observatory using a chemistry-coupled regional forecasting model. *Monthly Notices of the Royal Astronomical Society*, 530(2), 1414–1423. https://doi.org/10.1093/mnras/stae727

- Bran, S. H., Macatangay, R., Surapipith, V., et al. (2022). *Atmospheric Research*, 277, 106303. https://doi.org/10.1016/j.atmosres.2022.106303

These studies demonstrate the model's reliability in capturing air quality dynamics, optical turbulence, and regional transport processes over northern Thailand. Nonetheless, we will revise the manuscript to include a brief discussion confirming the WRF-Chem model's validation status, along with appropriate citations, to clarify its suitability as a reference for comparison in this study (added to the abstract, introduction, methodology, Figure 3 caption, conclusion and references section):

Abstract:

Accurate determination of the planetary boundary layer (PBL) height, mixing layer height (MLH), and aerosol layer top (ALT) is critical for air quality and climate studies, particularly in regions with complex aerosol dynamics such as Chiang Mai, northern Thailand. This study presents a novel LiDAR-based methodology that incorporates a temperature-dependent, dynamic maximum analysis altitude (MAA) to decouple these layers, addressing the limitations of conventional algorithms like the Haar Wavelet Covariance Transform (WCT). Traditional fixed-altitude approaches often misclassify the ALT as the PBL height—especially during nighttime or transition periods—leading to significant overestimations. By dynamically adjusting the MAA based on surface temperature variations, the proposed approach more effectively distinguishes the PBL from residual aerosol layers and cloud interference. Validation using radiosonde data and comparison with WRF-Chem simulations demonstrate strong agreement, with LiDAR-derived PBL heights exhibiting improved diurnal resolution and accuracy. However, model simulations tend to overestimate the PBL height during periods of elevated aerosol loading, underscoring the need for improved aerosol−radiation interaction parameterizations. The WRF-Chem model used in this study has been previously validated for northern Thailand and provides a robust benchmark for PBL comparison. This analysis highlights seasonal variations in agreement metrics and supports the integration of thermodynamic and aerosol observations for enhanced boundary layer characterization. The framework developed here offers a reliable tool for advancing air quality forecasting, pollutant transport analysis, and LiDAR-based remote sensing applications across Southeast Asia.

This study focuses on enhancing LiDAR-based boundary layer characterization by refining the detection of PBL height, MLH, and ALT. Traditional algorithms, such as the Haar Wavelet Covariance Transform (WCT), often misclassify the ALT as the PBL height, especially at night or during transitional periods when aerosol gradients are less distinct. Clouds and other atmospheric complexities make these measurements more challenging. By integrating normalized relative backscatter (NRB) profiles with dynamic thermodynamic adjustments, this approach addresses ambiguities in traditional methods and improves the reliability of boundary layer determinations. The novel method developed in this study was validated using radiosonde measurements and compared against WRF-Chem simulations. To support model−observation comparisons, we use a WRF-Chem configuration that has been previously validated under similar regional conditions in northern Thailand for surface pollutant distributions, boundary layer dynamics, and optical turbulence (Bran et al., 2022; Macatangay et al., 2024; Bran et al., 2024), confirming its suitability as a benchmark.

The WRF-Chem simulations used in this study have been previously validated under similar regional conditions in northern Thailand. Prior work has demonstrated the model's reliability in capturing key atmospheric dynamics, including surface $PM_{2.5}$ distributions, optical turbulence, and boundary layer processes. Notably, the model has been successfully applied in the following studies: Bran et al. (2022), Macatangay et al. (2024), and Bran et al. (2024). These validations support the robustness of WRF-Chem for use as a benchmark in our comparison with LiDAR-derived PBL heights.

Figure 3. Comparison of planetary boundary layer (PBL) height estimates from LiDAR retrievals (red line) and WRF-Chem forecasts (black line) at NARIT AstroPark, Chiang Mai, Thailand, for (a) December 2023, (b) January 2024, and (c) February 2024 (left column). The middle column shows time series of differences between WRF-Chem and LiDAR estimates (WRF − LiDAR), while the right column presents scatter plots with correlation coefficients (r), root mean square error (RMSE), percentage RMSE (%RMSE), and number of matched data

points (n). The WRF-Chem model configuration used in this comparison has been previously validated under similar regional conditions in northern Thailand, supporting its use as a reference for PBL height estimation.

Conclusion (2nd paragraph):

This study highlights the complexities of aerosol layering and PBL identification in regions with high aerosol loading, such as Chiang Mai, where seasonal forest fires and agricultural burning contribute to significant atmospheric pollution. By enhancing the accuracy of LiDAR-based PBL height estimations, the research provides critical insights for improving air quality modeling and understanding pollutant transport under complex meteorological conditions. The results demonstrate a generally consistent relationship between LiDAR-derived and model-based PBL height estimates, with seasonal variations in agreement metrics. The WRF-Chem model configuration, previously validated over northern Thailand for air quality, turbulence, and PBL structure, continues to perform reliably as a comparison benchmark. These findings support the value of integrating high-resolution LiDAR retrievals with regional models to evaluate boundary layer processes and diagnose modeling uncertainties across different seasons.